# Chest radiograph findings in children aged 2–59 months hospitalised with community-acquired pneumonia, prior to the introduction of pneumococcal conjugate vaccine in India: a prospective multisite observational study

Shally Awasthi [1], Tuhina Rastogi,[1] Neha Mishra,[1] Abhishek Chauhan,[2] Namita Mohindra,[3] Ram Chandra Shukla,[4] Monika Agarwal,[5] Chandra Mani Pandey,[6] Neera Kohli,[7] CAP Study Group[8]

For numbered affiliations see end of article.

**Correspondence to**
Professor Shally Awasthi;
shally07@gmail.com

## ABSTRACT

**Objectives** The current study was a hospital-based surveillance of cases hospitalised with WHO-defined community-acquired pneumonia in children aged 2–59 months, to assess the radiological abnormalities in chest X-rays and to identify the demographic and clinical correlates of specific radiological abnormalities, in residents of prespecified districts of Uttar Pradesh and Bihar, India.

**Design** Prospective, active, hospital-based surveillance.

**Setting** Multisite study conducted in a network of 117 secondary/tertiary care hospitals in four districts of Uttar Pradesh and Bihar, India.

**Participants** Included were children aged 2–59 months, hospitalised with community-acquired pneumonia, residing in the project district, with duration of illness <14 days and who had not been hospitalised elsewhere for this episode nor had been recruited previously.

**Main outcome measure** Concordant radiological abnormalities in the chest X-rays.

**Results** From January 2015 to April 2017, 3214 cases were recruited and in 99.40% (3195/3214) chest X-rays were available, among which 88.54% (2829/3195) were interpretable. Relevant radiological abnormalities were found in 34.53% (977/2829, 95% CI 32.78 to 36.28). These were primary end point pneumonia alone or with other infiltrates in 22.44% (635/2829, 95% CI 20.90% to 23.98%) and other infiltrates in 12.09% (342/2829; 95% CI 10.88% to 13.29%). There was a statistically significant interdistrict variation in radiological abnormalities. Statistically significantly higher proportion of abnormal chest X-rays were found in girls, those with weight-for-age z-score ≤−3SD, longer duration of fever, pallor and with exposure to biomass fuel.

**Conclusions** Among hospitalised cases of community-acquired pneumonia, almost one-third children had abnormal chest radiographs, which were higher in females, malnourished children and those with longer illnesses; and an intra-district variation was observed.

## Strengths and limitations of this study

► Prospective, multisite study recruiting cases from a large hospital surveillance network established for the project in four districts in two states of India that have high under-5 mortality rates.

► WHO definition of clinical pneumonia was used for identifying hospitalised cases for generalisability.

► Radiological abnormalities were interpreted by a panel of three independent, trained radiologists outside the surveillance network, blinded to each other as well as clinical features of the case.

► Since pre-existing X-ray machines were used, there were variations in the quality of images, which was, however, minimised by digitising them centrally.

► Since data of clinical examination were abstracted from hospital records, interobserver variation in documentation was possible.

## INTRODUCTION

Community-acquired pneumonia (CAP) is the leading cause of death in young children worldwide. Globally, pneumonia accounts for 16% of deaths in children under 5 years of age, which translates into almost 1 million deaths annually, with 0.9 million deaths reported in 2016.[1 2] Most deaths due to pneumonia occur in low-income and middle-income countries, particularly in sub-Saharan Africa and South Asia.[2 3] In India, there were approximately 0.44 million under-5 deaths due to CAP in the year 2015.[4]

CAP could be of either viral or bacterial aetiology.[5–7] In young children, bacteria associated with pneumonia are predominantly *Streptococcus pneumoniae* and *Haemophilus influenzae* type b, while viruses are respiratory

syncytial virus and influenza A or B.[6] However, aetiology varies from country to country and also across different time periods. To reduce the incidence of bacterial pneumonia, vaccination against *Haemophilus influenzae* type b is already under the national immunisation programme of India since 2011. Thereafter, WHO introduced pneumococcal conjugate vaccine (PCV) in countries, such as India, with high child mortality rates.[8] Consequently, PCV-13 was launched in May 2017 under the national immunisation programme of five Indian states (Uttar Pradesh, Bihar, Rajasthan, Madhya Pradesh and Himachal Pradesh) in a phased manner.[9] It is expected to be rolled out in other parts of the country in the near future.

Differentiating bacterial from viral aetiology of CAP based on clinical features or investigations remains difficult.[7 10 11] Therefore, several PCV probe trials have used radiographically confirmed end point pneumonia to be a surrogate marker of bacterial aetiology and hence used this as an outcome measure for vaccine efficacy. This approach has been endorsed by WHO.[12–14]

The current study was conducted to assess the radiological abnormalities in chest X-rays (CXRs) and to identify the demographic and clinical correlates of specific radiological abnormalities in children aged 2–59 months, hospitalised with WHO-defined CAP, residing in pre-specified districts of Uttar Pradesh and Bihar.

## METHODS
### Study design and setting
This was a prospective, multisite observational study conducted in the northern Indian states of Uttar Pradesh and Bihar. Uttar Pradesh is the first most populated and Bihar third most populated state of the country.[15 16] This study was conducted in Lucknow and Etawah districts of Uttar Pradesh and Patna and Darbhanga districts of Bihar, India. In Lucknow district, 66.2% population is urban and in Patna district 43.07%.[15 16] In contrast, only 22.3% population of Etawah district and 9.74% population of Darbhanga district is urban.[15 16] All four project districts have high infant and child mortality rates.[15–17] Infant mortality rate per 1000 live births of Lucknow district is 44, Etawah district 56, Patna district 31 and Darbhanga district 44, all being higher than the national average 41.[15–17] Similarly, under-5 mortality rates per 1000 live births of districts included in this study are above the national average 50, being 58 for Lucknow, 85 for Etawah, 46 for Patna and 77 for Darbhanga.[15–17]

### Study population
This study was conducted after obtaining institutional ethical clearance from all four participating academic institutions, one in each district. Each institution then established a prospective, active, hospital-based surveillance system for this study.[17 18] After obtaining written informed consent from the private hospital management and district administration for public hospitals, included in the surveillance were 117 public and private hospitals of four study districts which provided either secondary or tertiary level care to admitted children.

Surveillance officers of the project visited these hospitals every 48–72 hours to screen and recruit eligible cases. In between the scheduled visits they telephonically contacted the hospitals daily to inquire about hospitalisation of any potentially eligible case and made additional visits, if required. All children between the ages of 2 – 59 months, hospitalised in network hospitals with history of fast breathing with/without chest in-drawing were screened.[18]

Included were children hospitalised with symptoms of WHO-defined CAP and residing in the project district.[18] WHO-defined CAP was categorised into pneumonia and severe pneumonia. Fast breathing ≥50 breaths/min in a child aged 2–11 months and ≥40 breaths/min in a child aged 12–59 months, with or without chest in-drawing was categorised as 'pneumonia'.[19] Cough or difficulty in breathing plus at least one of the following: (a) oxygen saturation <90% or central cyanosis or (b) severe respiratory distress (eg, grunting, very severe chest in-drawing) or (c) signs of pneumonia with a general danger sign (inability to breast feed or drink, lethargy or reduced level of consciousness, convulsions) was categorised as 'severe pneumonia'.[19] Excluded were children with cough for ≥14 days or those who had been hospitalised in last 14 days.[18]

### Sample size
We assumed that the incidence of radiological pneumonia is 3/100 child years of observations. Then for a margin of error of 1.5/100 child years of observation, incidence of pneumonia in the community of 20/100 child years of observation, alpha level of 0.05 and power of 90% when the estimated population of children under 5 years of age in Lucknow district[20] is 750 000; 693 cases had to be included per district.

### Data collection
Data were collected by surveillance officers hired for the project at each of the four district sites. They had postgraduate degree in social sciences and at least 10 years experience in community-based health research. After recruitment, they were imparted 6-day centralised training on project procedures and logistics. Classroom as well as practical skills training in real-life setting was given by the coordinating centre in Lucknow. Pre-tests and post-tests were conducted to ascertain knowledge. Skills acquired by them were assessed during field observations. The coordinating centre provided annual refresher training to the surveillance officers from all four district sites in Lucknow. This was done to ensure quality of data collected.

After obtaining written, informed consent of the caregivers, data were collected through face-to-face interviews with caregivers, as well as by abstraction from hospital records. Sociodemographic data obtained by interviewing caregivers were: child's age, gender, residence,

birth order, immunisation status, current breastfeeding status, parental education and occupation, smoking status of parents, family type, housing infrastructure, use of biomass fuel, etc. Caregivers were also asked about the symptoms of disease and its duration in days.

Clinical data, recorded by pre-existing hospital staff at the time of hospitalisation, were abstracted by surveillance officers. Data were collected on anthropometry (weight and height), fever (axillary temperature ≥37.5°C), oxygen saturation by pulse oxymetry where done, pallor, central cyanosis, signs of pneumonia along with general danger sign and vital signs (heart rate and respiratory rate). Presence of auscultatory wheeze was abstracted or inquired from the treating clinician. In case information on a clinical variable was missing in the medical chart, the surveillance officers contacted the clinician and obtained the same. Thus, there were no missing data for clinical variables reported in this manuscript.

At the hospitals, clinicians generally used Integrated Management of Childhood Illness definitions[21] to identify pallor, cyanosis, wheeze on auscultation and general danger sign as it is incorporated in their medical undergraduate training. Most clinicians of public health sector had also received a formal in-service training on Integrated Management of Childhood Illness.[21] Clinical outcome (survival or mortality) was noted from the hospital records on follow-up.[17 18]

### Chest X-ray image acquisition and archiving

CXR (posterior-anterior view) was done on the advice of treating physician. These CXRs were obtained by the surveillance officers at the time of recruitment. CXRs were either analogue or digital. In case of digital CXRs, second copy was obtained where possible. If only single analogue image was available, then the hardcopy of CXR was obtained from the caregiver after the child was discharged. If the caregiver was not ready to give the hardcopy of CXR (in <1% cases), image of the same was captured by surveillance officers using 16 megapixel cell phone camera and portable CXR view box.

CXRs of recruited cases were subsequently scanned and converted into digital format using a diagnostic quality film image digitizer (Microteck International Limited, Medi 6000 plus).[22] These were archived for web-based radiological interpretation. Digital images were stored in JPEG format at 300 dpi resolution. Each CXR file was anonymised and given a unique identification number. Digital CXRs were uploaded on customised online data management software.

### Interpretation of radiological images

A panel of radiologists was constituted for standardised interpretations of CXRs. Four radiologists were part of this panel, one of whom was project co-investigator-Radiology (NK). All radiologists are faculty in medical teaching institutes and also look after pediatric radiology. They have >15 years of experience in interpreting paediatric CXRs.

Radiologists were trained according to the methodology developed by Department of Immunization, Vaccines and Biologicals of WHO for research purpose.[11] An international WHO-certified trainer from the International Centre for Diarrhoeal Disease Research, Bangladesh imparted 2-day in-house training to the radiologists. The objectives of this training were to standardise interpretation and coding of CXRs, to develop a CXR reporting form (online supplementary appendix S1) and to provide training on web-based CXR retrieval and reporting system. During the training, 210 CXRs of the WHO data set were used. For assessing post-training concordance, another set of 48 CXRs was provided for interpretation to individual radiologists. Post-test agreement with WHO findings was about 80%. Interobserver variation was about 25% and was for only minor interpretations such as quality of film, end point infiltrates, etc. Repeat training was conducted on an additional set of 44 CXRs provided by WHO to ensure standardisation in interpretation. Thereafter, concordance achieved by the radiologists was reviewed quarterly by the study arbitrator. Radiologists met annually to review key concepts and discuss challenges faced in interpreting CXRs.

After training, radiologists independently reviewed CXRs and registered their findings in an online standardised chest radiograph interpretation form (online supplementary appendix S1). For optimal viewing of CXRs, all radiologists used similar workstations. Specifications were provided for the computer monitor and hardware to be used. It was ensured that computer monitors had the correct brightness and contrast adjustment for optimal viewing.

During online evaluation, radiologists reported the quality of film as '*interpretable*' or '*un-interpretable*'. Furthermore, they categorised '*interpretable*' CXRs as either 'adequate/optimal' which allowed for confident interpretation of consolidation and pleural effusion as well as other infiltrates or 'suboptimal' which allowed interpretation of only consolidation and pleural effusion, but not of other infiltrates. In '*un-interpretable*' CXRs, no comment was possible for radiological abnormality such as consolidation, pleural effusion or other infiltrates.[12]

After interpreting film quality, radiologists evaluated interpretable CXRs for abnormal radiological findings. For each CXR evaluated, radiological abnormality could be presence of consolidation, other infiltrates or pleural effusion. '*Consolidation*' was defined as a dense or confluent opacity that occupied a portion or whole of a lobe or the entire lung that may or may not contain air bronchograms. '*Other infiltrates*' were defined as linear and patchy opacities (interstitial infiltrate) in a lacy pattern, featuring peri-bronchial thickening and multiple areas of atelectasis, also including minor patchy infiltrates that were not of sufficient magnitude to constitute end point consolidation, and small areas of atelectasis which may be difficult to distinguish from consolidation. '*Pleural effusion*' was defined as the fluid in the lateral pleural space between the lung and chest wall that was spatially associated with a

pulmonary parenchymal infiltrate (including 'other infiltrates') or had obliterated enough of the hemithorax to obscure any infiltrates. In most cases, this was to be seen at the costo-phrenic angle or as a layer of fluid adjacent to the lateral chest wall and this does not include fluid seen in the horizontal or oblique fissures.[12] Primary end point pneumonia (PEP) for research purpose was the presence of consolidation or pleural effusion which could be with or without other infiltrates.

Final conclusions were categorised as: (a) *abnormal* when it was 'PEP only' or 'other infiltrates only' or 'both PEP and other infiltrates' and (b) *normal* when no abnormal findings were seen.[12]

Data manager checked for inconsistencies and completeness after online evaluation of CXRs by individual radiologists. Thereafter, CXRs with concordant and discordant interpretations were identified. Interpretations were considered concordant when there was an agreement between two or more radiologists on final conclusions and discordant if all the three radiologists disagreed. Discordant interpretations were forwarded to the study arbitrator (NK). Arbitrator assessed discordant CXRs online and her interpretation was taken as final.

## Data management and statistical analysis

Clinical data of hospital surveillance network were entered online in customised software. Primary entry was done by the four participating sites. Secondary data entry was done by the coordinating site in separate customised software. Anonymised CXRs were uploaded on customised software. Each of the three panellists assessed the CXRs online, blind to peer assessments as well as clinical features of the case. CXR assessment data were downloaded from the online software in MS Access database.

Exploratory data analysis was performed for detection of outlier and missing observations for all the variables. Un-interpretable CXRs were not analysed. Among interpretable CXRs, concordant radiological abnormalities were taken as final. Weight-for-age (WAZ) z-score of each child was calculated using WHO Anthro Survey Analyser.[23] Weight of 7.59% (215/2829) children was missing. Missing weight was estimated using regression-based imputation technique.[24] Kappa statistics was performed for agreement analysis among radiologists for CXRs findings. Statistical analysis was performed using SPSS V.22.0 (Chicago, Illinois, USA).[25] A p value of <0.05 was taken as statistically significant using a two-tailed distribution.

Univariate analysis was performed to evaluate heterogeneity, stratified by four participating districts for sociodemographic variables such as child's age, gender, residence, birth order, immunisation status, current breastfeeding status, parental education and occupation, smoking status of parents, family type, housing infrastructure, use of biomass fuel and for clinical variables such as weight, height, duration of fever and oxygen saturation.

We report proportions of radiological abnormalities among children hospitalised for CAP by four districts. Univariate analysis was performed to assess association of

sociodemographic variables and clinical signs of CAP with radiological abnormalities. Student's t-test was used for continuous variables and $\chi^2$ test was used for categorical variables. Analysis of variance was used to test the significance of continuous variables when there were more than two groups. Multivariate unconditional logistic regression was performed to find association of presence of various radiological abnormalities with other variables that had univariate association with radiological abnormalities (p≤0.2) and/or were clinically meaningful, controlling for district of residence.

We developed four models in which the dependent (outcome) were different CXR findings and these were as follows:

Model I: abnormal versus normal.

Model II: PEP alone or with other infiltrates versus normal.

Model III: PEP alone versus normal.

Model IV: other infiltrates only versus normal.

Independent variables were the same in all the four models. These were participating districts, age, gender, use of biomass fuel, symptoms of CAP such as duration of illness, presence of wheeze on auscultation, pallor, vomiting everything and malnutrition status of the case (WAZ ≤−2 SD (malnourished) and WAZ ≤−3 SD (severely malnourished)).

## Patient and public involvement in research

Patients or public were not involved in the development of research question, study design or conducting the research. Reporting of this research conforms to the guidelines for Strengthening the Reporting of Observational Studies in Epidemiology.[26]

## RESULTS

From January 2015 to April 2017, 3290 cases were screened in hospital surveillance network of four districts. Out of these, 3214 were eligible and consenting for inclusion (figure 1). Among them, in 3195 (99.40%) CXR was done and only in 19 (1.0%) cases CXR was not done. However, only 88.54% (2829/3195) CXRs were interpretable and remaining 11.45% (366/3195) were un-interpretable. In cases with interpretable CXRs, 99.11% (2804/2829) had 'severe pneumonia' as per the WHO criteria.[19]

Concordance among ≥2 radiologists on final conclusion of CXRs findings was 86.0%. Kappa statistics was calculated for agreement of CXRs findings between reader 1 versus reader 2 ($K_1$=0.31), reader 2 versus reader 3 ($K_2$=0.46) and reader 3 versus reader 1 ($K_3$=0.42). Among interpretable CXRs, 22.44% (635/2829) cases had PEP alone or with other infiltrates, 12.09% (342/2829) had other infiltrates only and 65.46% (1852/2829) were normal (figure 1).

Table 1 shows univariate distribution of sociodemographic and clinical variables among hospitalised cases across participating districts. A variation was observed in sociodemographic variables such as place of residence,

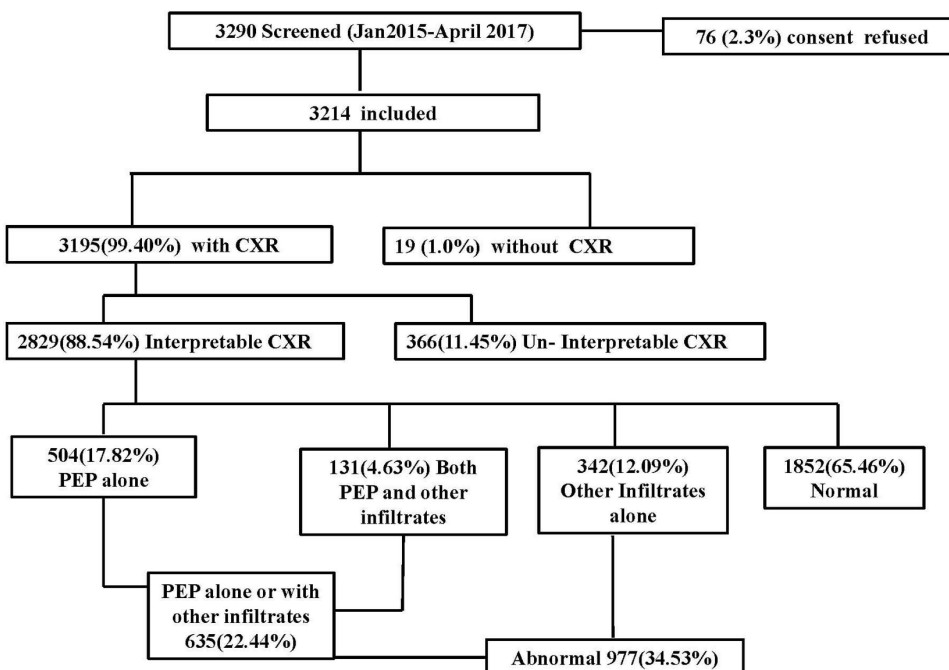

**Figure 1** Flow diagram of cases of community-acquired pneumonia recruited from participating districts before the introduction of pneumococcal conjugate vaccine (January 2015–April 2017). CXR, chest X-ray; PEP, primary end point pneumonia.

type of house, type of family, maternal and paternal education and occupation, use of biomass fuel and parental smoking status across the four districts. We also report clinical variables of recruited cases across the four districts in table 1. Among those where pulse-oximetry was done, the proportion of cases with oxygen saturation <90% was found to be different across four districts.

Among those with interpretable CXRs, 34.5% (977/2829) had radiological abnormalities, which were PEP alone or with other infiltrates in 64.9% (635/977) and other infiltrates in 35.1% (342/977) (table 2). In the same table, we report these findings by district as well as sociodemographical and clinical associates of normal versus abnormal CXR as well among those with abnormal CXRs, in those with PEP alone or with other infiltrates versus other infiltrates only.

We observed statistically significant district-wise heterogeneity in radiological abnormalities (table 2). We found higher proportion of radiological abnormalities as well as PEP alone or with other infiltrates in Patna and Lucknow districts, and lower proportion in Etawah and Darbhanga districts. Statistically significant higher proportion of females hospitalised for CAP had radiologically abnormal CXR (table 2). Likewise, statistically significantly higher proportion of abnormal CXR findings were reported in hospitalised cases who had symptoms of fever, pallor, wheezing on auscultation or were malnourished (table 2). Among cases with abnormal CXRs, statistically significantly higher proportion of cases with other infiltrates had wheezing on auscultation.

Table 3 describes four multivariate unconditional logistic regression models to find associates of various abnormal CXR findings. After controlling for age, gender, symptoms of pneumonia, duration of illness, biomass fuel and malnutrition status of cases, statistically significant district-wise heterogeneity remained in the first three models. Model I, which compared abnormal versus normal CXRs, model II, which compared CXRs with PEP alone or with other infiltrates versus normal and model III, which compared CXRs with PEP alone versus normal, had similar associates for radiological abnormalities whereas model IV, which compared CXRs with other infiltrates only versus normal, was different. Across all the four models, female gender and those with severe malnutrition had statistically significantly higher risk for having abnormal CXRs. A higher risk of radiological abnormalities was also observed in those children with longer duration of illness.

## DISCUSSION

This active, prospective, hospital-based surveillance study was conducted to assess radiological abnormalities in CXRs and to identify the demographic and clinical correlates of specific radiological abnormalities in children aged 2–59 months, hospitalised with WHO defined CAP, residing in pre-specified districts of Uttar Pradesh and Bihar. The study was conducted from January 2015 to April 2017, prior to the introduction of PCV in the national immunisation programme of the Government of India.[9]

In our study, among interpretable CXRs, we found that 22.44% (635/2829) cases had PEP alone or with infiltrates, 12.09% (342/2829) had other infiltrates only and

**Table 1** Distribution of sociodemographic and clinical variables among hospitalised children for participating districts (January 2015–April 2017)

| Characteristics | Lucknow | Etawah | Patna | Darbhanga | Total |
|---|---|---|---|---|---|
| Sociodemographic characteristics | n=1025 (%) | n=389 (%) | n=744 (%) | n=671 (%) | n=2829 (%) |
| **Gender** | | | | | |
| Male | 659 (64.29) | 287 (73.78) | 557 (74.87) | 502 (74.81) | 2005 (70.87) |
| **Place of residence** | | | | | |
| Rural | 195 (19.02) | 279 (71.72) | 304 (40.86) | 614 (91.51) | 1392 (49.20) |
| Urban | 830 (80.98) | 110 (28.28) | 440 (59.14) | 57 (8.49) | 1437 (50.80) |
| **Family type** | | | | | |
| Joint | 688 (67.12) | 360 (92.54) | 707 (95.03) | 383 (57.08) | 2138 (75.57) |
| Nuclear | 337 (32.88) | 29 (7.46) | 37 (4.97) | 287 (42.77) | 690 (24.39) |
| **House type** | | | | | |
| Mud | 64 (6.24) | 54 (13.88) | 123 (16.53) | 374 (55.74) | 615 (21.74) |
| Bricks | 854 (83.32) | 256 (65.81) | 453 (60.89) | 85 (12.67) | 1648 (58.25) |
| Combined | 107 (10.4) | 79 (20.31) | 168 (22.58) | 212 (31.59) | 566 (20.01) |
| **Mother's education** | | | | | |
| No formal education | 203 (19.80) | 56 (14.40) | 328 (44.09) | 496 (73.92) | 1083 (38.28) |
| Class I–V | 108 (10.54) | 28 (7.20) | 82 (11.02) | 38 (5.66) | 256 (9.05) |
| Class VI–XII | 379 (36.98) | 176 (45.24) | 243 (33.66) | 112 (16.69) | 910 (32.17) |
| Graduate/Postgraduate | 335 (32.68) | 129 (33.16) | 91 (12.23) | 25 (3.73) | 580 (20.50) |
| **Father's education** | | | | | |
| No formal education | 167 (16.29) | 29 (7.46) | 153 (20.56) | 345 (51.42) | 694 (24.53) |
| Class I–V | 85 (8.29) | 19 (4.88) | 91 (12.23) | 82 (12.22) | 277 (9.79) |
| Class VI–XII | 437 (42.63) | 206 (52.96) | 328 (44.09) | 205 (30.55) | 1176 (41.57) |
| Graduate/Postgraduate | 336 (32.78) | 135 (34.70) | 172 (23.12) | 39 (5.81) | 682 (24.11) |
| **Birth order** | | | | | |
| First | 435 (42.44) | 187 (48.07) | 315 (42.34) | 192 (28.61) | 1129 (39.91) |
| Second | 343 (33.46) | 120 (30.85) | 235 (31.59) | 258 (38.45) | 956 (33.79) |
| Third | 153 (14.93) | 47 (12.08) | 129 (17.34) | 137 (20.42) | 466 (16.47) |
| More than third | 93 (9.07) | 35 (9.00) | 62 (8.33) | 83 (12.37) | 273 (9.65) |
| **Immunisation status** | | | | | |
| Complete for age | 792 (77.27) | 300 (77.12) | 711 (95.56) | 544 (81.07) | 2347 (82.96) |
| Incomplete for age | 220 (21.46) | 84 (21.59) | 25 (3.36) | 126 (18.78) | 455 (16.08) |
| Unimmunised | 13 (1.27) | 5 (1.29) | 8 (1.08) | 1 (0.15) | 27 (0.95) |
| **Currently breast feeding** | | | | | |
| Yes | 653 (63.71) | 256 (65.81) | 589 (79.17) | 537 (80.03) | 2035 (71.93) |
| No | 372 (36.29) | 133 (34.19) | 155 (20.83) | 134 (19.97) | 794 (28.07) |
| **Father's occupation** | | | | | |
| Unemployed | 13 (1.27) | 20 (5.14) | 27 (3.63) | 63 (9.39) | 123 (4.35) |
| Daily wages | 329 (32.10) | 81 (20.82) | 165 (22.18) | 474 (70.64) | 1049 (37.08) |
| Salaried/Professional | 397 (38.73) | 104 (26.74) | 245 (32.93) | 55 (8.20) | 801 (28.31) |
| Self- employed | 286 (27.90) | 184 (47.30) | 307 (41.26) | 79 (11.77) | 856 (30.26) |
| **Mother's occupation** | | | | | |
| Home maker | 961 (93.76) | 376 (96.66) | 701 (94.22) | 484 (72.13) | 2522 (89.15) |
| Daily wages | 17 (1.66) | 3 (0.77) | 17 (2.28) | 171 (25.48) | 208 (7.35) |
| Salaried/Professional | 47 (4.59) | 9 (2.31) | 18 (2.42) | 7 (1.04) | 81 (2.86) |
| Self-employed | 0 (0.0) | 1 (0.26) | 8 (1.08) | 9 (1.34) | 18 (0.64) |
| **Biomass fuel** | | | | | |

Continued

| Table 1 Continued | | | | | |
|---|---|---|---|---|---|
| Characteristics | Lucknow | Etawah | Patna | Darbhanga | Total |
| **Sociodemographic characteristics** | **n=1025 (%)** | **n=389 (%)** | **n=744 (%)** | **n=671 (%)** | **n=2829 (%)** |
| Yes | 211 (20.59) | 245 (62.98) | 263 (35.35) | 609 (90.76) | 1328 (46.94) |
| No | 814 (79.41) | 144 (37.02) | 481 (64.65) | 62 (9.24) | 1501 (53.06) |
| **Smoking status—father** | | | | | |
| Yes | 152 (14.83) | 45 (11.57) | 56 (7.53) | 59 (8.79) | 312 (11.03) |
| No | 873 (85.17) | 344 (88.43) | 688 (92.47) | 612 (91.21) | 2517 (88.97) |
| **Indoor smoking—father** | | | | | |
| Yes | 83 (8.10) | 21 (5.40) | 16 (2.15) | 43 (6.41) | 163 (5.76) |
| No | 942 (91.90) | 368 (91.60) | 728 (97.85) | 628 (93.59) | 2666 (94.24) |
| **Smoking status— family member** | | | | | |
| Yes | 129 (12.59) | 55 (14.14) | 45 (6.05) | 102 (15.20) | 331 (11.70) |
| No | 896 (87.41) | 334 (85.86) | 699 (93.95) | 569 (84.80) | 2498 (83.30) |
| **Indoor smoking— family member** | | | | | |
| Yes | 84 (8.20) | 27 (6.94) | 27 (3.63) | 94 (14.01) | 232 (8.20) |
| No | 941 (91.80) | 362 (93.06) | 717 (96.37) | 577 (85.99) | 2597 (91.80) |
| **Clinical variables at the time of admission at hospital** | **n Mean±SD** | **n Mean±SD** | **n Mean±SD** | **n Mean±SD** | **n Mean±SD** |
| Age (months) | 1025 14.53±13.88 | 389 10.69±10.95 | 744 10.26±11.35 | 671 12.30±13.29 | 2829 12.35±12.85 |
| Height (cm) | 303 68.61±13.78 | 324 70.66±13.75 | 34 64.38±10.25 | 266 70.46±12.14 | 927 69.70±13.26 |
| Weight (kg) | 1025 7.96±2.97 | 389 7.34±2.73 | 744 7.11±2.78 | 671 7.78±2.93 | 2829 7.61±2.90 |
| Fever duration (days) | 929 4.46±2.71 | 321 3.59±2.37 | 689 4.25±2.52 | 569 3.54±2.47 | 2508 4.08±2.59 |
| **Respiratory rate** | | | | | |
| Respiratory rate (2–11 months) | 602 53.38±14.05 | 272 60.87±9.60 | 540 53.82±10.16 | 451 60.78±7.26 | 1864 56.37±11.49 |
| Respiratory rate (12–59 months) | 423 47.75±14.17 | 117 53.22±13.17 | 204 45.59±10.11 | 220 58.03±6.83 | 964 50.30±12.76 |
| Oxygen saturation done (n, %) | 528 (51.51) | 343 (88.17) | 236 (34.25) | 319 (56.06) | 1426 (50.40) |
| Oxygen saturation <90% (n, %) | 61 (11.53) | 57 (16.61) | 49 (20.76) | 43 (13.47) | 210 (14.72) |
| Grunting (n, %) | 461 (44.98) | 353 (90.75) | 687 (92.34) | 649 (96.72) | 2150 (76.00) |
| Very severe chest in-drawing (n, %) | 953 (92.97) | 352 (90.49) | 739 (99.33) | 651 (97.02) | 2695 (95.26) |
| **Signs of pneumonia with a general danger sign** | | | | | |
| Lethargy or reduced level of consciousness (n, %) | 423 (41.27) | 259 (66.58) | 6 (0.81) | 412 (61.40) | 1100 (38.88) |
| Inability to breast feed or drink (n, %) | 291 (28.39) | 259 (66.58) | 75 (10.08) | 312 (46.50) | 937 (33.12) |
| Convulsions (n, %) | 16 (1.56) | 19 (4.58) | 13 (1.75) | 100 (14.90) | 148 (5.23) |
| Central cyanosis (n,%) | 15 (1.46) | 7 (1.80) | 26 (3.49) | 14 (2.09) | 62 (2.19) |

65.46% (1852/2829) had normal findings. Our study used WHO case definition for CAP.[19] A panel of three trained radiologists interpreted CXRs, adopting WHO recommended methodology.[11 12] These make our study methodology robust and results generalisable.

There were 88.54% (2829/3195) interpretable CXRs in the current study. This is similar to 83% (3587/3973) interpretable CXRs reported by Pneumonia Aetiology Research for Child Health (PERCH) study conducted on 4232 children (1–59 months) in nine sites in seven countries.[27] Consistent with PERCH findings, a vaccine probe trial conducted in Gambia found proportion of interpretable CXRs among unvaccinated cases of pneumonia to be 84.32% (242/287).[28]

**Table 2** Distribution of sociodemographic and clinical factors by chest radiograph findings among hospitalised children from January 2015 to April 2017

| | | Interpretable chest X-rays | | | Abnormal chest X-rays | | |
| --- | --- | --- | --- | --- | --- | --- | --- |
| | n=2829 | Normal 1852 n (%) | Abnormal 977 n (%) | P value | PEP* alone or with other infiltrates 635 n (%) | Other infiltrates 342 n (%) | P value |
| **Participating site (row %)** | | | | | | | |
| Lucknow | 1025 | 636 (62.05) | 389 (37.95) | <0.0001 | 282 (72.49) | 107 (27.51) | <0.0001 |
| Etawah | 389 | 275 (70.69) | 114 (29.31) | | 73 (64.04) | 41 (35.96) | |
| Patna | 744 | 457 (61.42) | 287 (38.58) | | 184 (64.11) | 103 (35.89) | |
| Darbhanga | 671 | 484 (72.13) | 187 (27.87) | | 96 (51.34) | 91 (48.66) | |
| **Sociodemographic and clinical factors (column %)** | | | | | | | |
| **Age group (months)** | | | | | | | |
| 2–11 | 1865 | 1223 (66.04) | 642 (65.71) | 0.86 | 409 (64.41) | 233 (68.13) | 0.26 |
| 12–59 | 964 | 629 (33.96) | 335 (34.29) | | 226 (35.59) | 109 (31.87) | |
| **Gender** | | | | | | | |
| Male | 2005 | 1354 (73.11) | 651 (66.63) | <0.0001 | 426 (67.09) | 225 (65.79) | 0.72 |
| Female | 824 | 498 (26.89) | 326 (33.37) | | 209 (32.91) | 117 (34.21) | |
| **Place of residence** | | | | | | | |
| Rural | 1392 | 921 (49.73) | 471 (48.21) | 0.44 | 299 (47.09) | 172 (50.29) | 0.34 |
| Urban | 1437 | 931 (50.27) | 506 (51.79) | | 336 (52.91) | 170 (49.71) | |
| **Biomass fuel** | | | | | | | |
| Yes | 1501 | 867 (46.81) | 461 (47.19) | 0.44 | 294 (42.30) | 167 (48.83) | 0.24 |
| No | 1328 | 985 (53.19) | 516 (52.81) | | 341 (53.70) | 175 (51.17) | |
| **Immunisation status** | | | | | | | |
| Complete for age | 2347 | 1546 (83.48) | 801 (81.99) | 0.32 | 516 (81.26) | 285 (83.33) | 0.54 |
| Incomplete | 482 | 306 (16.52) | 176 (18.01) | | 119 (18.74) | 57 (16.67) | |
| **Clinical features** | | | | | | | |
| Fever | 2499 | 1616 (87.26) | 883 (90.38) | 0.014 | 575 (90.55) | 308 (90.06) | 0.82 |
| Pallor | 764 | 465 (25.11) | 299 (30.60) | 0.002 | 200 (31.50) | 99 (28.95) | 0.41 |
| Wheeze on auscultation | 2054 | 1377 (74.35) | 677 (69.29) | 0.005 | 415 (65.35) | 262 (76.61) | 0.0003 |
| Duration of illness fever (days) (n, mean±SD) | 2499 | 1611, 3.91±2.51 | 888, 4.40±2.70 | <0.0001 | 577, 4.57±2.82 | 342, 4.08±2.44 | 0.011 |
| **Respiratory rate** | | | | | | | |

Continued

**Table 2** Continued

| | Interpretable chest X-rays | | | Abnormal chest X-rays | | |
| --- | --- | --- | --- | --- | --- | --- |
| | | Normal 1852 n (%) | Abnormal 977 n (%) | P value | PEP* alone or with other infiltrates 635 n (%) | Other infiltrates 342 n (%) | P value |
| | n=2829 | Normal 1852 n (%) | Abnormal 977 n (%) | P value | PEP* alone or with other infiltrates 635 n (%) | Other infiltrates 342 n (%) | P value |
| Respiratory rate (2–11 months) (n, mean±SD) | 1865 | 1243 55.52±11.29 | 642 57.99±11.70 | <0.0001 | 409 58.12±11.88 | 233 57.74±11.40 | 0.69 |
| Respiratory rate (12–59 months) (n, mean±SD) | 964 | 629 49.78±12.41 | 335 51.28±13.37 | 0.08 | 226 51.35±13.31 | 109 51.12±13.35 | 0.88 |
| Fast breathing for age (2–11 months) | 1735 | 1130 (61.02) | 605 (61.92) | 0.11 | 384 (60.47) | 221 (64.62) | 0.69 |
| Fast breathing for age (12–59 months) | 862 | 562 (30.35) | 300 (30.71) | 0.92 | 204 (32.13) | 96 (28.07) | 0.53 |
| **Signs of pneumonia with a general danger sign n (%)** | | | | | | | |
| Lethargy or reduced level of consciousness | 1101 | 732 (39.52) | 369 (37.77) | 0.39 | 247 (38.90) | 122 (35.67) | 0.33 |
| Inability to breast feed or drink | 937 | 612 (33.05) | 325 (33.27) | 0.46 | 211 (33.23) | 114 (33.33) | 0.97 |
| Convulsions | 148 | 98 (5.29) | 50 (5.12) | 0.93 | 33 (5.20) | 17 (4.97) | 0.87 |
| Central cyanosis | 62 | 39 (2.11) | 23 (2.35) | 0.34 | 16 (2.52) | 7 (2.05) | 0.64 |
| **Malnutrition status** | | | | | | | |
| Normal* | 1880 | 1293 (69.82) | 587 (60.08) | <0.0001 | 367 (57.80) | 220 (64.33) | 0.06 |
| Malnutrition* | 517 | 333 (17.98) | 184 (18.83) | | 122 (19.21) | 62 (18.13) | |
| Severe malnutrition* | 432 | 226 (12.20) | 206 (21.08) | | 146 (22.99) | 60 (17.54) | |

*Normal: weight-of-age z-score >−2 SD; malnutrition: weight-for-age z-score ≤−2 SD and severe: malnutrition weight-for-age z-score ≤−3 SD.

There have been several studies in the past two decades, which have reported CXRs findings in hospitalised cases of paediatric CAP. Almost all of these were conducted before the introduction of PCV in their respective regions. A small prospective study conducted in Ethiopia reported radiological abnormality in CXRs in 48.3% (95% CI 39.49 to 57.22) among 122 children aged 3 months to 14 years with clinically diagnosed WHO-defined severe pneumonia.[29] Similar findings were reported from the Gambian vaccine probe trial where the proportion of radiological abnormality was 45% (95% CI 43.35 to 46.46) among unvaccinated hospitalised cases of clinical pneumonia.[28] Likewise, PERCH study found that 54% (95% CI 52.31 to 55.57) of CXRs among cases of CAP were abnormal.[27] In all of these studies, proportion of cases with abnormal CXRs is higher than 34.5% (95% CI 32.8 to 36.3) found by us in the current study. However, our findings are similar to PERCH rural study site of Matlab, Bangladesh that reported radiological abnormality in 35.3% (95% CI 29.77 to 40.85) CXRs of hospitalised cases of CAP.[27] Another PERCH urban site of Dhaka, Bangladesh reported 63.10% (95% CI 56.18 to 70.02) cases with abnormal CXRs.[27] In our study, radiological abnormalities in CXRs were higher in cases from largely urban districts of Patna and Lucknow compared with rural districts of Darbhanga and Etawah. This is consistent with rural-urban differences in Bangladesh sites of PERCH.[27] Variation in CXR findings among cases of CAP may be due to place of residence, infecting organism, immune response of patient and prior duration of disease.

In 2016, WHO's Department of Immunisation, Vaccines and Biologicals standardised the categorisation of radiological pneumonia and established that PEP can be taken as a good surrogate marker of bacterial pneumonia in epidemiological and vaccine efficacy studies.[11] In our study, 22.44% (95% CI 20.90 to 23.98) CXRs had PEP alone or with other infiltrates and hence were probably bacterial in aetiology. This is similar to findings of PERCH study that reported PEP alone or with other infiltrates in 27% (95% CI 25.50 to 28.40) hospitalised cases of CAP.[27] However, a study conducted in Gambia reported that 45% (95% CI 43.35 to 46.46) non-vaccinated children had PEP and/or other infiltrates,[28] which is higher than that found by us or the PERCH study. PEP in CXR has

**Table 3** Independent associations between chest radiograph findings and demographic and clinical variables among hospitalised children of community-acquired pneumonia, using unconditional logistic regression

| Variables | Model I Abnormal/Normal[ref] | | Model II PEP alone or with other infiltrates/Normal[ref] | | Model III PEP alone/Normal[ref] | | Model IV Other infiltrates/Normal[ref] | |
|---|---|---|---|---|---|---|---|---|
| | Adjusted OR (95% CI) | P value | Adjusted OR (95% CI) | P value | Adjusted OR (95% CI) | P value | Adjusted OR (95% CI) | P value |
| **Districts** | | | | | | | | |
| Lucknow vs others | 1.58 (1.20 to 2.10) | <0.0001 | 2.07 (1.48 to 2.89) | <0.0001 | 2.20 (1.52 to 3.19) | <0.0001 | 0.98 (0.65 to 1.47) | 0.93 |
| Etawah vs others | 1.22 (0.88 to 1.70) | 0.23 | 1.30 (0.87 to 1.95) | 0.19 | 1.49 (0.95 to 2.30) | 0.07 | 1.17 (0.74 to 1.87) | 0.50 |
| Patna vs others | 1.67 (1.27 to 2.20) | <0.0001 | 1.89 (1.36 to 2.64) | <0.0001 | 2.25 (1.56 to 3.24) | <0.0001 | 1.39 (0.95 to 2.07) | 0.09 |
| **Age group (months)** | | | | | | | | |
| 2–11[ref] | | | | | | | | |
| 12–59 | 0.92 (0.77 to 1.10) | 0.34 | 0.95 (0.77 to 1.17) | 0.62 | 1.03 (0.82 to 1.29) | 0.79 | 0.86 (0.66 to 1.13) | 0.27 |
| **Gender** | | | | | | | | |
| Male[Ref] | | | | | | | | |
| Female | 1.39 (1.16 to 1.66) | <0.0001 | 1.34 (1.08 to 1.65) | 0.008 | 1.28 (1.01 to 1.61) | 0.03 | 1.48 (1.14 to 1.92) | 0.004 |
| **Symptoms of pneumonia*** | | | | | | | | |
| Wheezing | 0.83 (0.68 to 1.01) | 0.06 | 0.72 (0.57 to 0.90) | 0.005 | 0.75 (0.59 to 0.96) | 0.02 | 1.14 (0.83 to 1.55) | 0.42 |
| Pallor | 1.30 (1.08 to 1.58) | 0.006 | 1.28 (1.03 to 1.60) | 0.02 | 1.22 (0.95 to 1.55) | 0.12 | 1.34 (1.01 to 1.77) | 0.04 |
| Vomiting everything | 0.90 (0.75 to 1.09) | 0.28 | 0.80 (0.64 to 0.99) | 0.04 | 0.78 (0.62 to 1.01) | 0.05 | 1.09 (0.83 to 1.08) | 0.51 |
| Duration of illness, fever (days) | 1.06 (1.04 to 1.09) | <0.0001 | 1.08 (1.04 to 1.12) | <0.0001 | 1.08 (1.04 to 1.12) | <0.0001 | 1.03 (0.98 to 1.48) | 0.24 |
| Biomass fuel | 1.28 (1.05 to 1.57) | 0.02 | 1.39 (1.10 to 1.76) | 0.006 | 1.40 (1.14 to 1.88) | 0.003 | 1.08 (0.79 to 1.45) | 0.64 |
| **Malnutrition status** | | | | | | | | |
| Normal†[ref] | | | | | | | | |

| Variables | Model I Abnormal/Normal[ref] | | Model II PEP alone or with other infiltrates/Normal[ref] | | Model III PEP alone/Normal[ref] | | Model IV Other infiltrates/Normal[ref] | |
|---|---|---|---|---|---|---|---|---|
| | Adjusted OR (95% CI) | P value | Adjusted OR (95% CI) | P value | Adjusted OR (95% CI) | P value | Adjusted OR (95% CI) | P value |
| Malnutrition† | 1.18 (0.93 to 1.45) | 0.15 | 1.17 (0.91 to 1.52) | 0.23 | 1.17 (0.88 to 1.55) | 0.27 | 1.12 (0.82 to 1.52) | 0.47 |
| Severe malnutrition† | 1.65 (1.31 to 2.09) | <0.0001 | 1.82 (1.34 to 2.36) | <0.0001 | 1.87 (1.41 to 2.47) | <0.0001 | 1.62 (1.71 to 2.23) | 0.003 |

**Table 3** Continued

*No signs of pneumonia taken as a reference.
†Normal: weight-for-age z-score >−2 SD; malnutrition: weight-for-age z-score ≤−2 SD; severe malnutrition: weight-for-age z-score ≤−3 SD.
PEP, primary end point pneumonia; Ref, reference category.

been associated with increased risk of treatment failure (p=0.002), increased length of hospitalisation (p=0.0003) and more days of respiratory support (p=0.002) in Botswana when compared with cases reporting no significant pathology' on CXRs.[30]

In our study, female gender (p<0.001) was at the higher risk of developing radiological abnormalities compared with males (table 3). The results are in concordance with a hospital-based case-control study carried out in Brazil that reported male gender as a protective factor against pneumonia (OR=0.53; 95% CI 0.39 to 0.72).[31] However, a study in Mozambique, Africa reported that male gender was not significantly associated with presence of radiological abnormalities (OR=0.77; 95% CI 0.56 to 1.05) in children (0–59 months) suffering from severe pneumonia.[32] In contrast, a Gambian study reported male preponderance for all pneumonia that was most marked for those whose CXRs showed 'other infiltrates/abnormalities'.[28]

In our study, we observed differential care-seeking by gender for CAP in all four project districts. Although females admitted with CAP were at higher risk of having radiological abnormalities, lower proportions were hospitalised. Gender inequality in healthcare seeking for females is common in India, as in other South Asian countries.[33 34] Since there is no healthcare financing or provision of health insurance in India, in case of severe illness, parents are less likely to incur out-of-pocket expenditure or incur debts to pay expenses on medical treatment of their daughters as compared with sons.[35] Another Indian study found that male children were five times more likely to be taken early for medical care and three times more likely to be seen by qualified medical doctors compared with female children.[36] We also found that large proportion of hospitalised cases of pneumonia were from urban areas, as there is poor healthcare seeking from rural areas.[37]

A systematic review with meta-analysis conducted in 2014 suggests that no one clinical feature is sufficient on its own to diagnose radiological pneumonia.[38] However, other sociodemographic and clinical correlates of abnormal CXRs found by us in model I (abnormal vs normal), which increased the risk of radiological abnormalities were presence of pallor, severe malnutrition, longer duration of illness and exposure to biomass fuel. In low-income and middle-income countries, exposure to biomass fuel used for cooking has been reported as an important risk factor for CAP.[39] In rural India, majority of the households use biomass fuel like firewood, dung cakes and wood for cooking.[40] Young children are at risk of adverse effects of exposure to biomass fuel as either the households have no separate cooking space or have poor ventilation and sometimes young children stay with their mother while she cooks.

Other correlates of PEP/radiological pneumonia, which were more likely to be bacterial in aetiology, as found in model II, which compared PEP alone or with other infiltrates versus normal, and model III, which compared PEP alone versus normal (table 3), besides

those found in model I, were presence of vomiting everything and wheeze on auscultation, both of which were found to be protective. These symptoms/signs are more often reported in viral pneumonia.[41] Correlates of radiological abnormalities of 'other infiltrates' (model IV which compared other infiltrates with normal), which increased the risk, were again female gender, pallor and severe malnutrition. Hence, it is difficult to attribute radiological findings of other infiltrates to either bacterial or viral aetiology.

Based on our study, almost two-third hospitalised cases of CAP had normal CXRs and could be perhaps of viral aetiology. This is supported by a recent study that reported 61.4% (95% CI 57.3 to 65.6) cases to be viral.[41] One-third of cases of CAP had abnormal CXRs and thus were more likely to be bacterial in aetiology.

In India, 13-valent PCV has been introduced in May 2017. A three-dose schedule is followed with two primary and one booster, at 6 weeks, 14 weeks and 9 months of age, respectively. PCV 13 provides coverage against 13 serotypes (1, 3, 4, 5, 6A, 6B, 7F, 9V, 14, 18C, 19A, 19F and 23F).[42] Several studies have assessed serotype distribution of pneumococcal disease among children in India. A study conducted in Vellore, India, found that the most common serotypes causing invasive infections among under-5 children were 14, 19F, 5, 6A and 6B, all of which are covered by the 13-valent PCV.[43] Another population-based surveillance study conducted in rural Bangladesh found that the most common serotypes of *S. pneumoniae* were 1, 5, 14, 18C and 19A and 38, which caused invasive disease and all but one were covered by the 13-valent vaccine.[44] A systematic review and meta-analysis of data collected on invasive pneumococcal disease serotypes from under-5 children during the pre-PCV period (between 1980 and 2007) found that serotypes included in both the 10-valent and 13-valent PCVs accounted for 10 million cases and 600 000 deaths worldwide.[45]

Several strengths of the study are worth noting. This was an active, prospective, multisite study where recruitments were done from a large hospital surveillance network established especially for the study in four districts of two Indian states that have high under-5 mortality rates. Standard WHO definition was used to identify hospitalised cases of CAP. Radiological abnormalities were interpreted by a panel of three trained radiologists at locations out of the surveillance network, blinded to each other as well as clinical features of the case. Despite these strengths, our study findings have certain limitations. First, in our study, pre-existing x-rays machines which were not of uniform specification were used. This might have caused variation in quality of CXR images, although this error was minimised by digitising the CXR images centrally. Second, in our study, clinical data collection was recorded by clinicians in the network hospitals and there could be observer bias. This could also have lead to possibly over reporting of presence of wheezing. In this study, we have not collected information on use of antibiotic prior to hospitalisations; as such information is not available

reliably. However, in another study done in one of the network hospitals of Lucknow in the recent past, it was found that 70.5% children tested positive for antibiotics on urine examination.[46] Prior use of antibiotics could have possibly led to underestimation of radiological pneumonia. We also observed that pulse oxymetry was not routinely done in the network hospitals. This could have an impact on the case management but would not have affected the radiological findings of CXRs.

## CONCLUSION

Among hospitalised cases of CAP, almost one-third of children had abnormal chest radiographs of which about two-thirds had abnormalities related to possible bacterial aetiology (*S. pneumoniae*). Hence, the introduction of pneumococcal vaccination is likely to reduce the burden of childhood pneumonia in India. Since the study was done prior to the introduction of PCV in India, continued surveillance will be required to assess the impact of PCV on radiological findings in cases admitted with CAP. The impact of introduction of PCV in the national immunisation programme on under-5 mortality rate and burden of CAP needs to be assessed too.

Author affiliations
[1]Department of Pediatrics, King George's Medical University, Lucknow, India
[2]Department of Radio-diagnosis, Dr. Ram Manohar Lohia Institute of Medical Sciences, Lucknow, India
[3]Department of Radio-diagnosis, Sanjay Gandhi Post Graduate Institute of Medical Sciences, Lucknow, India
[4]Department of Radio-diagnosis, Institute of Medical Sciences, Banaras Hindu University, Varanasi, India
[5]Department of Community Medicine and Public Health, King George's Medical University, Lucknow, India
[6]Department of Biostatistics & Health Informatics, Sanjay Gandhi Post Graduate Institute of Medical Sciences, Lucknow, India
[7]Department of Radio-diagnosis, King George's Medical University, Lucknow, India
[8]CAP Study Group, Lucknow, India

Acknowledgements The authors would like to thank the state governments of Uttar Pradesh and Bihar for giving permission to conduct the study. The authors would like to thank the participating tertiary and secondary care hospitals for their contribution in this study. The authors would like to thank Dr Kamrun Nahar from the International Centre for Diarrhoeal Disease Research, Bangladesh for training the panel of study radiologists. Members of the Lucknow Community-Acquired Pneumonia (CAP) Group: King George's Medical University (KGMU), Lucknow, India: Professor Shally Awasthi, Department of Pediatrics, Professor Neera Kohli, Department of Radio-diagnosis, Professor Monika Agarwal, Department of Community Medicine and Public Health; Mr Puneet Dhasmana, Project Staff; Patna Medical College and Hospital, Patna, Bihar: Professor Neelam Verma and Professor Chandra Bhushan Kumar, Department of Pediatrics; Darbhanga Medical College and Hospital, Darbhanga, Bihar, India: Professor Chittaranjan Roy, Department of Community Medicine, Professor Kripanath Mishra, Department of Pediatrics; Uttar Pradesh University of Medical Sciences, Etawah, Uttar Pradesh, India: Professor Pankaj Kumar Jain, Department of Community Medicine, Professor Rajesh Yadav, Department of Pediatrics.

Collaborators None.

Contributors The study was conceived and designed by SA. CAP Study Group performed data acquisition. CMP and NeM conducted the statistical analysis of the data. The paper was written by SA, TR, MA and CMP. AC, NaM, RCS and NK interpreted chest X-rays. All authors were involved with drafting and revising the work and approved the final submission.

**Funding** The study was supported by Bill & Melinda Gates Foundation (https://www.gatesfoundation.org/) via Grant No: OPP1118005.

**Disclaimer** Funding agency had no role in the design of study and collection, analysis and interpretation of data or in writing the manuscript.

**Competing interests** None declared.

**Patient and public involvement** Patients and/or the public were not involved in the design, conduct, reporting or dissemination plans of this research.

**Patient consent for publication** Not required.

**Ethics approval** The Institutional Ethics Committee of King George's Medical University (Lucknow), The Uttar Pradesh University of Medical Sciences (Etawah), Patna Medical College and Hospital (Patna) and Darbhanga Medical College and Hospital (Darbhanga) gave ethical approval for conduct of study.

**Provenance and peer review** Not commissioned; externally peer reviewed.

**Data availability statement** Data are available on reasonable request. The data contained within this study can be obtained by writing to the corresponding author (SA) at email: shally07@gmail.com.

**Author note** Members of the Lucknow CAP Group are given in the Acknowledgement.

**ORCID iD**
Shally Awasthi http://orcid.org/0000-0003-1254-9802

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
