## [Reviewer comments · BMJ Open]

ARTICLE DETAILS

TITLE (PROVISIONAL)	Chest Radiograph Findings in children aged 2-59 months hospitalized with Community-Acquired Pneumonia, prior to the introduction of Pneumococcal Conjugate Vaccine in India- A Prospective Multisite Observational Study
AUTHORS	Awasthi, Shally; Rastogi, Tuhina; Mishra, Neha; Chauhan, Abhishek; Mohindra, Namita; Shukla, Ram; Agrawal, Monika; Pandey, CM; Kohli, Neera; Study Group, CAP

VERSION 1 – REVIEW

REVIEWER	Eric McCollum Johns Hopkins, United States
REVIEW RETURNED	28-Oct-2019

GENERAL COMMENTS	This is a review of the manuscript "Chest Radiograph Findings in children aged 2-59 months hospitalized with Community-Acquired Pneumonia, prior to the introduction of Pneumococcal Conjugate Vaccine in India" by Awasthi and colleagues. In general this is a potentially nice contribution to the literature. I do, however, have issues with the methodology of the study as outlined below. In addition, the writing will need to be improved for readability prior to publication. 1) Children with possible radiographic pneumonia were screened at hospitals but the clinical pneumonia case definition applied to the study population was appropriate to the community and not hospital. The authors used the WHO "integrated Community Case Management" criteria and not WHO "Integrated Management of Childhood Illness (IMCI)" or WHO "Pocketbook" criteria. This has implications for the generalizability of the study findings and needs clarification. Preferably, the authors should apply a more appropriate definition (either IMCI or Pocketbook) to their study population. iCCM would be appropriate if children were screened by community health workers from the community. 2) Data collection was done by "trained surveillance officers" but no information is given on these professionals. Detailed information should be provided on exactly who these people are, their level of training etc. 3) Data source. As written it is currently unclear to me where the data is coming from as some clinical data came from the surveillance officers and some data came from the clinical records. 4) Who trained the radiologists to interpret the CXRs? The training and supervision of the radiologists and their interpretations needs
--

	detailed explanation. Were radiologists required to pass a certification test in order to read CXRs for the project (as would be standard)? Was their performance tracked during the readings and was any remediation conducted? 5) Who was the "study arbitrator" that interpreted discordant CXRs? What was this person's training and expertise? 6) Oxygen saturation is one of the most important clinical variables and while the authors indicate that this variable was collected it was ultimately not offered to the models as an independent variable. Why? In addition, <90% is the currently recommended WHO threshold for hypoxemia but the authors used a <=92% threshold? Why? Should be justified. 7) Rhonchi: Many settings use different terminology for lung sounds and rhonchi can mean different things in different settings (may indicate wheeze but its original definition was crackles). The authors should define what rhonchi means and where it was identified on the patient (chest?) and the data source as I presume the surveillance officers did not auscultate the chest of children. 8) Other infiltrate: Other infiltrate is notoriously a heterogenous category from the WHO CXR definitions and has little clinical relevance. For table 2 why did the authors not run a comparison of PEP alone or with other infiltrate vs normal? To me this would be the most important comparison but does not seem to have been done. 9) Table 2: Table 2 lacks many respiratory signs that one would expect to have been included in the dataset. I have listed some: -respiratory rate -fast breathing for age -lower chest wall indrawing -vomiting everything (authors have included "vomiting", which is very different and not a danger sign) -stridor in a calm child -oxygen saturation <90% 10) Table 2: please clarify whether the categories of "malnutrition" and "severe malnutrition" are intended to overlap? What does "difficulty in breathing" mean? How was this determined (caregiver report or observation)? Difficulty in breathing is not considered a danger sign at hospitals (per WHO). 11) Table 3: Authors need to describe the variables used in the models. All variables in Table 3 were offered to all models? Why wasn't oxygen saturation used? Was oxygen saturation recorded while the child was breathing in room air, or could it also be obtained while on oxygen? Did the authors account for this in the analyses? 12) pallor: Do the authors mean severe palmar pallor? Just pallor? How was this determined? 13) fever: what is the definition of fever? Axillary temp? Core temp? Threshold? 14) The discussion section needs additional development to improve readability and flow.
--	--

REVIEWER	Kerry-Ann O'Grady
-----------------	-------------------

	Queensland University of Technology Australia
REVIEW RETURNED	02-Dec-2019

GENERAL COMMENTS	Overall this is a useful paper for those countries in which PCV has not been introduced. That the findings are consistent with most PEP studies adds strength to the study. It is one of the few such studies that report the outcomes of WHO standardisation with respect to handling of images and assessing concordance amongst xray readers. My main comments are as follows:  1. Why did you not capture wheeze from clinical records and report on this as this is now considered important, particularly given the prevalence of wheeze amongst children with CXR abnormalities and your high proportion of cases that are likely to viral. If you did collect wheeze, it would be useful to add to the paper and compare wheeze prevalence amongst your radiological endpoints. 2. Were the radiologists paediatric radiologists? If so, please add this to your manuscript. 3. I think there is a lack of explanation of some of your findings such as female gender, urban vs rural differences as these do differ across various studies. Some context around these factors in the regions in which the study was done would be helpful. For example, are these known to be associated with healthcare seeking behaviours and/or propensity to admit to hospital? 4. I note that antibiotic use prior to hospitalisation was not reported which may be an important influence on the findings given a lot of pneumonia is managed in the community. Would the authors kindly confirm as to whether or not that was collected, and if it wasn't address the potential impact on your findings this may have. 5. What did you define as fever? Please specify this in the text and in your tables 6. Table 1. SpO2 < 92%. I think this would be more useful reported as a proportion of cases with that on admission rather than the mean values. 7. Who extracted the clinical data from medical records and was this done in a standardised manner and independent of those accessing xrays? 8. Some data on known prevalence of S. pneumoniae vaccine-type serotypes in the region would be useful if that is available. If not, are there data on this from similar other regions in India. 9. I note childcare attendance was not collected. Given its importance in respiratory infections, it would be useful to have data on that from the study if available. If not, then clarifying why it was not collected would be good (ie....is childcare infrequently used in these regions). 10. Overall, the manuscript has multiple grammatical errors with respect to written English. I started listed them but there were too many. The paper needs to reviewed/amended by an English editor before publication.
--

	11. Unless I have missed something, the STROBE checklist was not provided with the manuscript.
--	--

VERSION 1 – AUTHOR RESPONSE

Reviewer: 1: Reviewer Name: Eric McCollum Institution and Country: Johns Hopkins, United States	
1)	Children with possible radiographic pneumonia were screened at hospitals but the clinical pneumonia case definition applied to the study population was appropriate to the community and not hospital. The authors used the WHO "integrated Community Case Management" criteria and not WHO "Integrated Management of Childhood Illness (IMCI)" or WHO "Pocketbook" criteria. This has implications for the generalizability of the study findings and needs clarification. Preferably, the authors should apply a more appropriate definition (either IMCI or Pocketbook) to their study population. iCCM would be appropriate if children were screened by community health workers from the community
	We thank the reviewer for the comment. The patients were hospitalized and categorized according to Integrated Management of Childhood Illness criteria. We apologize for giving incorrect reference. The correct reference has now been given as reference number 19: World Health Organization. (2013). Pocket book of hospital care for children: guidelines for the management of common childhood illnesses, 2nd ed.. World Health Organization. https://apps.who.int/iris/handle/10665/81170 The following has been added in the manuscript: "WHO has developed guidelines for hospital-based management of common childhood illness such as pneumonia¹⁹. According to these guidelines, fast breathing ≥ 50 breaths/minute in a child aged 2–11 months and ≥ 40 breaths/minute in a child aged 12-59 months along with chest indrawing was categorized as having `pneumonia`¹⁹. A child presenting with cough or difficulty in breathing with: (a) oxygen saturation $< 90\%$ or central cyanosis (b) severe respiratory distress (e.g. grunting, very severe chest indrawing) and (c) signs of pneumonia with a general danger sign (inability to breastfeed or drink, lethargy or reduced level of consciousness, convulsions) was categorized as having `severe pneumonia`¹⁹" [Page 6, Line 52-60]
2)	Data collection was done by "trained surveillance officers" but no information is given on these professionals. Detailed information should be provided on exactly who these people are, their level of training etc.
	We have added the following in the manuscript : "Data was collected by surveillance officers who had postgraduate degree in social sciences and almost 10 years experience in community based health research. After recruitment, they were imparted six-day centralized training on project procedures and logistics. Class-room as well as practical skills-training was given by the coordinating centre in Lucknow. Pre and post tests were conducted to ascertain knowledge and skills acquired by them through the training to ensure quality in data collection. The coordinating centre provided annual refresher trainings to the surveillance officers from all four sites in Lucknow." [Page 7, Line No. 69-75]
3)	Data source. As written it is
	We have clarified and have added the following in the

	currently unclear to me where the data is coming from as some clinical data came from the surveillance officers and some data came from the clinical records.	manuscript : “After obtaining written, informed consent of the caregivers, data was collected through face-to-face interviews with them as well as by abstraction from hospital records. Socio-demographic data, obtained by interviewing caregivers, was: child’s age, gender, residence, birth order, immunization status, current breastfeeding status, parental education and occupation, smoking status of parents, family type, housing infrastructure, use of biomass fuel etc. Caregivers were also asked about the symptoms of disease and its duration in days. Clinical data, recorded by pre-existing hospital staff at the time of hospitalization, was abstracted. Where available, data was collected on anthropometry (weight and height), fever (axillary temperature $\geq 37.5^{\circ}\text{C}$), oxygen saturation by pulse oxymetry, pallor, central cyanosis, and danger signs of pneumonia and vital signs (heart rate and respiratory rate). Presence of wheezing on auscultation of chest was abstracted, when recorded. At the hospitals, clinicians generally used Integrated Management of Childhood Illness (IMCI) definitions²⁰ to identify pallor, cyanosis, wheezing on auscultation and general danger sign as it is incorporated in their medical undergraduate training. Most doctors of public health sector also receive a formal in-service training on IMCI 20. Clinical outcome (survival or mortality) was noted from hospital records on follow up.^{17 18”} [Page 7-8, Line No. 77-93]
4)	 • Who trained the radiologists to interpret the CXRs? The training and supervision of the radiologists and their interpretations needs detailed explanation.  • Were radiologists required to pass a certification test in order to read CXRs for the project (as would be standard)? • Was their performance tracked during the readings and was any remediation conducted? 	We have added the following in the manuscript to respond to the query: “Radiologists were trained according to the methodology developed by Department of Immunization, Vaccines, and Biologicals of the WHO ¹¹. An international WHO-certified trainer from the International Centre for Diarrhoeal Disease Research, Bangladesh imparted a two-day in-house training to the radiologists. Training objective was to standardize interpretation and coding of CXRs, to develop a CXR reporting form [S1 Appendix] and to provide training on web-based CXR retrieval and reporting system. During the training, 210 CXRs of WHO data set were used. For assessing concordance post training, another set of 48 CXRs was provided for interpretation to individual radiologists. Post-test agreement with WHO findings was calculated, which was about 80%. Inter-observer variation was about 25% and was for minor interpretation like quality of film, end point infiltrates etc. Repeat training was conducted on an additional set of 44 CXRs provided by WHO to ensure standardization in interpretation. Thereafter, concordance achieved by the radiologists was reviewed quarterly by the study arbitrator. Radiologists met annually to review key concepts and

		discuss challenges faced in interpreting CXRs.” [Page 9, line No. 117-130]
5)	Who was the "study arbitrator" that interpreted discordant CXRs? What was this person's training and expertise?	Study arbitrator is an experienced radiologist and academician who has >30 years experience in interpreting pediatric radiology. Arbitrator is also a member of the World Health Organization Chest Radiography in Epidemiological Studies project and Executive council member of Indian Society of Pediatric Radiology. However, these details have not been added in the manuscript.
6)	 Oxygen saturation is one of the most important clinical variables and while the authors indicate that this variable was collected it was ultimately not offered to the models as an independent variable. Why? In addition, <90% is the currently recommended WHO threshold for hypoxemia but the authors used a <=92% threshold? Why? Should be justified. 	We agree with the reviewer that oxygen saturation is one of the most important clinical variable. In our study, oxygen saturation was collected from hospital records and was available in only half cases (1426/2829, 50.4%) cases and also collection of information on oxygen saturation by participating hospital was not a part of protocol. Since it was not possible to impute almost half the data, thus, it was not offered to the models as an independent variable. We have now used the currently recommended <90% WHO threshold for hypoxemia. This has been changed at the relevant places in the manuscript
7)	Rhonchi: Many settings use different terminology for lung sounds and rhonchi can mean different things in different settings (may indicate wheeze but its original definition was crackles). The authors should define what rhonchi means and where it was identified on the patient (chest?) and the data source as I presume the surveillance officers did not auscultate the chest of children.	We thank the reviewer for this comment. We agree with the reviewer that many settings use different terminology for lung sounds and rhonchi can mean different things in different settings and it may indicate wheeze We have replaced the word “ronchi” with “wheeze on auscultation” in our manuscript.  Information was abstracted from hospital records on “presence or absence of wheeze on auscultation”. The surveillance officers did not auscultate the chest of children but abstracted this information from hospital records, whenever available.
8)	Other infiltrate: Other infiltrate is notoriously a heterogenous category from the WHO CXR	We agree with the reviewer. We did run this model (comparison of PEP alone or with other infiltrate vs normal in Model 2). In the original manuscript, it is

	definitions and has little clinical relevance. For table 2 why did the authors not run a comparison of PEP alone or with other infiltrate vs normal? To me this would be the most important comparison but does not seem to have been done.	included as Table 3, Model 2. We have retained it in the revised version of the manuscript.
9)	Table 2: Table 2 lacks many respiratory signs that one would expect to have been included in the dataset. I have listed some: -respiratory rate -fast breathing for age -lower chest wall indrawing -vomiting everything (authors have included "vomiting", which is very different and not a danger sign) -stridor in a calm child -oxygen saturation <90%	Table 2: We have now added the following respiratory signs in Table 2  1. Respiratory rate: Added in Table 2 2. Fast breathing for age: Added in Table 2 3. Lower chest wall indrawing: We have mentioned in the methods section that child presenting with/without chest indrawing was part of inclusion criteria and hence data on it was not collected 4. Vomiting everything: In our study, we had collected data on the variable "Vomiting everything" as per IMNCI guidelines. Thus to give clarity, we have now correctly mentioned it as "Vomiting everything" in revised version. This has been changed at all places relevant places in the manuscript 5. Stridor in a calm child: This variable was not collected in our study 6. Oxygen saturation <90%: Added in Table 1
10)	Table 2:  • Please clarify whether the categories of "malnutrition" and "severe malnutrition" are intended to overlap? • What does "difficulty in breathing" mean? How was this determined (caregiver report or observation)? Difficulty in breathing is not considered a danger sign at hospitals (per WHO). 	 • Yes, we agree that they do overlap. However, we have described Malnutrition `Methods` section as "$WAZ \leq -2$ (malnourished)" and severe malnutrition as "$WAZ \leq -3$ (severely malnourished)" [Page 13, Line 217-218]. Thus severe malnutrition is a subset of malnutrition. • We agree with the reviewer that difficulty in breathing is not a general danger sign as per WHO and has therefore been removed from the table.

11)	Table 3: Authors need to describe the variables used in the models. All variables in Table 3 were offered to all models? Why wasn't oxygen saturation used? Was oxygen saturation recorded while the child was breathing in room air, or could it also be obtained while on oxygen? Did the authors account for this in the analyses?	We have described the variables in the manuscript. To provide further clarification, the following has been added in the manuscript "Where available, data was collected on anthropometry (weight and height), fever (axillary temperature $\geq 37.5^{\circ}\text{C}$), oxygen saturation by pulse oxymetry, pallor, central cyanosis, and danger signs of pneumonia and vital signs (heart rate and respiratory rate). Presence of wheezing on auscultation of chest was abstracted, when recorded. At the hospitals, clinicians generally used Integrated Management of Childhood Illness (IMCI) definitions²⁰ to identify pallor, cyanosis, wheeze on auscultation and general danger sign as it is incorporated in their medical undergraduate training. Most doctors of public health sector also receive a formal in-service training on IMCI²⁰." [Page 7-8, line No. 84-93]. For oxygen saturation related queries, kindly see our response to Point 6 above
12)	Pallor: Do the authors mean severe palmar pallor? Just pallor? How was this determined?	In our study, pallor was clinically reported by the clinicians of hospital network according to the Integrated Management of Childhood Illness guidelines (IMCI) guidelines. Clinicians, did not however, differentiate between `some palmer pallor` and `severe palmar pallor` while noting on hospital records and hence we could not further categorize it.
13	Fever: what is the definition of fever? Axillary temp? Core temp? Threshold?	Fever was taken as axillary temperature $\geq 37.5^{\circ}\text{C}$ in accordance with IMCI guidelines. This has been added in the manuscript. [Page 7, line No. 80].
14)	The discussion section needs additional development to improve readability and flow.	Discussion section has been further developed to improve readability and flow. New paragraphs have been added as per suggestions of both the reviewers A paragraph on Gender Differences in health care seeking has been added [Page 23-24, line No. 348-358] and another paragraph on prevalence of S. pneumoniae vaccine-type serotypes in the region [Page 25, line No. 383-395] has been added.
Reviewer: 2: Reviewer Name: Kerry-Ann O'Grady Institution and Country: Queensland University of Technology, Australia		
Overall this is a useful paper for those countries in which PCV has not been introduced. That the findings are consistent with most PEP studies adds strength to the study. It is one of the few such studies that report the outcomes of WHO standardization with respect to handling of images and assessing concordance amongst x-ray readers. My main comments are as follows		
1.	 Why did you not capture wheeze from clinical records and report on this as this is now considered important, particularly given the prevalence of wheeze amongst children with CXR abnormalities and your high proportion of cases that are likely 	We agree with the reviewer. What was captured in clinical record form by the clinicians was added sounds during expiration and this is categorically termed as wheezing. We have corrected this in the manuscript. We apologize for the error in the original version of the manuscript. Since we have now changed the terminology from `ronchi` to `wheezing`, we have compared prevalence of

	to viral.  If you did collect wheeze, it would be useful to add to the paper and compare wheeze prevalence amongst your radiological endpoints. 	wheezing among radiological end-points (Table 3)
2.	Were the radiologists paediatric radiologists? If so, please add this to your manuscript.	The following has been added in the manuscript in response to reviewers comments " All radiologists are faculty in medical teaching institutes and also look after pediatric radiology. They all have more than fifteen years experience in interpreting pediatric CXRs " [Page 9, line 114-115]
3.	I think there is a lack of explanation of some of your findings such as female gender, urban vs rural differences as these do differ across various studies. Some context around these factors in the regions in which the study was done would be helpful. For example, are these known to be associated with healthcare seeking behaviours and/or propensity to admit to hospital?	We completely agree with this comment. Health seeking behaviour varies by gender in India as in other South Asian countries. Also, place of residence impacts health seeking behavior. We have discussed these in the revised version of the manuscript. A new paragraph has been added as per suggestions of the reviewers [Page 23-24, line No. 348-358]
4.	I note that antibiotic use prior to hospitalization was not reported which may be an important influence on the findings given a lot of pneumonia is managed in the community. Would the authors kindly confirm as to whether or not that was collected, and if it wasn't address the potential impact on your findings this may have.	Antibiotic use prior to hospitalization was not collected in this study. This has been added in the Discussion Section in manuscript as follows: " We have not collected information on use of antibiotic prior to hospitalization; as such information is not available reliably. However, in another study, done in one of the network hospitals of Lucknow, in the recent past, it was found that 70.5% children tested positive for antibiotics on urine examination ⁴⁵ ." [Page 26, line No. 409-412]
5.	What did you define as fever? Please specify this in the text and in your tables	Fever was taken as axillary temperature $\geq 37.5^{\circ}\text{C}$ in accordance with `Integrated Management of Childhood Illness` guidelines. This has been added at relevant places in the manuscript.
6.	Table 1. SpO2 < 92%. I think this would be more useful reported as a proportion of cases with that on admission rather than the mean values.	In the revised version of manuscript, we have used the currently recommended <90% WHO threshold for hypoxemia. This has been changed at the relevant places in the manuscript. Also, we have now added oxygen saturation as proportion and have removed the Mean \pm SD values as suggested by reviewer (Table 1)
7.	Who extracted the clinical data from medical records and was this done in a standardized manner and	We have clarified and have added the following in the methods section of manuscript " Clinical data, recorded by pre-existing hospital staff at the time of hospitalization, was abstracted. " [Page 7, Line No. 78-79] Yes, data

	independent of those accessing x-rays?	collection was done on standardized data collection form.
8.	Some data on known prevalence of S. pneumoniae vaccine-type serotypes in the region would be useful if that is available. If not, are there data on this from similar other regions in India.	Discussion section has been further developed to improve readability and flow. New paragraphs have been added as per suggestions of both the reviewers. A paragraph on Gender Differences in health care seeking has been added A paragraph on Gender Differences in health care seeking has been added [Page 23-24, line No. 348-358] and another paragraph on prevalence of S. pneumoniae vaccine-type serotypes in the [Page 25, line No. 383-395] has been added.
9	I note childcare attendance was not collected. Given its importance in respiratory infections, it would be useful to have data on that from the study if available. If not, then clarifying why it was not collected would be good (ie...is childcare infrequently used in these regions).	In India, children are culturally not sent to childcare. Hence, we did not collect information on it.
10.	Overall, the manuscript has multiple grammatical errors with respect to written English. I started listed them but there were too many. The paper needs to reviewed/amended by an English editor before publication.	We have attempted to correct all the grammatical errors .
11.	Unless I have missed something, the STROBE checklist was not provided with the manuscript.	STROBE checklist for observational studies has been followed for this manuscript. The following has been added in this manuscript. "Reporting of this research conforms to the guidelines for Strengthening the Reporting of Observational Studies in Epidemiology (STROBE)²⁶". [Reference number 26]. STROBE checklist is also being attached as Annexure along with response to comments

VERSION 2 – REVIEW

REVIEWER	Eric McCollum Johns Hopkins University School of Medicine, USA
REVIEW RETURNED	04-Feb-2020

GENERAL COMMENTS	General comments: This is my second review of this manuscript. I still find the readability of the manuscript challenging and too lengthy. That said, I find the description of the surveillance system lacking and in need of much greater detail so that the reader can understand the findings. Currently I cannot. Abstract: The conclusion of the abstract is not aligned with the objectives of the abstract nor the stated purpose of the analysis as
---

described in the paper's introduction section. It would seem to me that the objectives were to assess whether primary endpoint pneumonia on chest radiography is common prior to introduction of PCV in order to assess whether PCV introduction is likely to be effective, and then to also look at any associations with CXR findings. As written the authors say nothing about Streptococcal pneumoniae and PCV until the conclusion, and so it comes across as unexpected.

Strengths and Limitations of the Study box:

The last bullet wording could be improved. I would expect that since the authors did not train any of the hospital physicians in WHO IMCI that there IS inter-observer variation present, and I would expect there to be missing clinical data (but there does not seem to be), this both are almost certain given they are present even when there is training done!

Introduction:

Page 4-5, lines 22-24: Primary endpoint pneumonia as defined by the WHO includes pleural effusion in their definition, so its not exactly "alveolar pneumonia." Throughout the manuscript the authors conflate radiographic pneumonia, alveolar pneumonia, and PEP. In my view its better for the authors to be specific with their wording so that its clear that they are using the WHO definition. It is also important to note somewhere that this approach is a research definition not intended for clinical use. These points are confused by many readers and its important to state this.

Methods:

Study population, page 6, line 45: Please state whether this hospital surveillance system was a passive or active. Its still not coming across clearly (at least to me) whether this system was one in which all children meeting IMCI pneumonia criteria had a CXR obtained or whether a CXR was obtained at the discretion of the treating physician. This is an important distinction. CXRs are not necessarily obtained on all children with clinical pneumonia, often only in children failing therapy or with more severe presentations, and so how these were obtained may introduce important bias into the study and should be addressed by the authors.

CXR acquisition, page 8, line 100: When a hardcopy of the CXR could not be obtained from the caregiver after hospital discharge, please explain how the image of the printed film was captured and whether a standard approach was applied. If not, this needs to be included as a limitation. Please also include in the results section how frequently this was an issue.

Page 11, lines 171-173: Please be specific regarding how the images were considered concordant or not. Was this only on PEP or no PEP? Or was it considered discordant if one reader said PEP only and another said PEP with OI?

Page 13, line 212: Please explicitly define Model 1. Please note Model 2 should say "other infiltrates", currently says "infiltrate."

Given this analysis is based on medical chart extraction and the physicians documenting clinical signs were not trained for this study I would assume that missing data was not insubstantial. For example, physicians may not always document "no wheezing" if wheezing was not found on their examination. Please clarify how

	such a scenario was handled. Was it assumed by the authors that lack of documentation of a clinical sign indicated that the sign was absent? If this approach was taken did the authors do sensitivity analyses assuming that the data was missing and did this alter the findings? Results: Page 14, line 228: I'm trying to understand why out of 3290 hospital cases screened the 3214 had WHO pneumonia? How many children were hospitalized during this time period and how were these 3290 hospital cases identified for screening? I'm not following the surveillance system. The fact that 3195 cases had CXRs suggests to me that CXRs were being obtained on nearly all children, regardless of clinical indication. Please clarify these points. Page 14, line 236: Clarify what is meant by concordance. Concordance on what? PEP vs no PEP? Or something else. Page 14, line 246: Why did the authors use $\leq 90\%$ for their hypoxemia threshold? The WHO uses $< 90\%$ in the 2014 IMCI chart booklet and in the hospital pocketbook. Page 15 & 16, table 1: I'm not sure the p values have any meaning in this table? Consider removing? Table 1: Please add the clinical variables used to classify children as having pneumonia, including respiratory rate, chest indrawing, WHO general danger signs. Include missingness if necessary. Table 2: I'm curious as to how there is no missing data of clinical variables given my understanding is that these data were extracted from the medical file. This would be unusual. Please clarify. Discussion: page 22, line 332: I'm not sure abbreviating SP is necessary, can write out. I also don't think that the WHO definition of PEP is intended to be considered a surrogate marker of streptococcal pneumoniae per say. PEP was intended to be a specific CXR endpoint enriched with bacterial etiology, of which some % was streptococcal pneumoniae. For example, I would not suggest that 22% of CXR PEP means that 22% of children with CXRs have streptococcal pneumoniae. Your current wording suggests this is the case and this is not correct. Page 24, line 373: I would remove "possibly due to SP" from this sentence.
--	---

REVIEWER	Kerry-Ann O'Grady Queensland University of Technology Australia
REVIEW RETURNED	13-Jan-2020

GENERAL COMMENTS	Thank you for addressing the reviewer comments and the manuscript is much improved. I think you need to have a sentence or two in the discussion (limitations section) about what impact the lack of SpO2 readings, the proportion of children diagnosed with wheeze and the lack of information of prior antibiotic use actually has on the findings (eg, under-estimate, overestimate, is disease
---

	likely to be bacterial etc) rather than just saying it was a limitation, however this may need to be an Editor decision based on existing length of the manuscript. There still needs some work done on English grammar but it is not an impediment to eventual publication.
--	--

VERSION 2 – AUTHOR RESPONSE

Reviewer Name: Eric McCollum Institution and Country: Johns Hopkins, United States. This is my second review of this manuscript. I still find the readability of the manuscript challenging and too lengthy		
1.	I find the description of the surveillance system lacking and in need of much greater detail so that the reader can understand the findings. Currently I cannot.	We have expanded the description of the surveillance system (Page 5 & 6, line 44-48). We have retained the reference (reference no. 18) to the protocol published earlier.
2.	Abstract: The conclusion of the abstract is not aligned with the objectives of the abstract nor the stated purpose of the analysis as described in the paper's introduction section. It would seem to me that the objectives were to assess whether primary endpoint pneumonia on chest radiography is common prior to introduction of PCV in order to assess whether PCV introduction is likely to be	We thank the reviewer for the taking the time out and reviewing the manuscript. We have done necessary correction in the `conclusion` section of the abstract. Earlier version: Among hospitalized cases of community-acquired pneumonia, almost one-third children had abnormal chest radiographs of which about two-thirds had abnormalities related with possible bacterial etiology (Streptococcus pneumoniae). Hence introduction of pneumococcal vaccination is likely to reduce burden of childhood pneumonia in India. Revised version: Among hospitalized cases of community-acquired pneumonia, almost one-third children had abnormal chest radiographs, which were higher in females, malnourished children and those with longer illnesses; and an intra-district variation was observed.

	effective, and then to also look at any associations with CXR findings. As written the authors say nothing about Streptococcal pneumoniae and PCV until the conclusion, and so it comes across as unexpected.	
3. 4	Strengths and Limitations of the Study box: The last bullet wording could be improved. I would expect that since the authors did not train any of the hospital physicians in WHO IMCI that there is inter-observer variation present, and I would expect there to be missing clinical data (but there does not seem to be), this both are almost certain given they are present even when there is training done!	The last bullet wording has been improved and it now reads as follows: Earlier version:  • Since the objective of the study was to assess the radiological abnormalities in chest X-rays of recruited cases, clinical data was recorded by pre-existing hospital staff, there could be some inter-observer variations. Revised version: Since data of clinical examination was abstracted from hospital records, it could have resulted in inter-observer variation.
4.	Introduction: Page 4-5, lines 22-24: Primary endpoint pneumonia as defined by the WHO includes pleural effusion in their definition, so its not exactly "alveolar	We have now removed the word alveolar pneumonia from page 4-5 line 22-23. At other places also, we have maintained uniformity in using the word "radiological pneumonia" We have mentioned in the manuscript that we have used WHO

	pneumonia" Throughout the manuscript the authors conflate radiographic pneumonia, alveolar pneumonia, and PEP. In my view its better for the authors to be specific with their wording so that its clear that they are using the WHO definition. It is also important to note somewhere that this approach is a research definition not intended for clinical use. These points are confused by many readers and it is important to state this.	definitions of radiological abnormalities found in chest x-rays for research purpose and these are not intended for clinical use (page 11, line 163-164). The following line has been added in the revised manuscript "Primary end point pneumonia for research purpose was the presence of consolidation or pleural effusion which could be with or without other infiltrates"
5.	Methods Study population, page 6, line 45: Please state whether this hospital surveillance system was a passive or active.	We have expanded the section on surveillance and clearly stated that it was a prospective, active, hospital-based surveillance system (Page 5 & 6, line 44-48).The following lines have been added in the revised manuscript (Page 5 & 6, line 44-48). A prospective, active, hospital-based surveillance system was established for this study^{17 18}. Included in the surveillance were 117 public and private hospitals of study districts which provided either secondary or tertiary level care to admitted children. Surveillance officers visited the hospital every 48-72 hours to recruit eligible cases. In between the visits they telephonically contacted the hospitals and made additional visits, if required. All children (2-59 months), hospitalized in network hospital between January 2015 to April 2017, with history of fast breathing with/without chest in-drawing were screened¹⁸. In our study, 99.1% of the cases had one or more severe presentations of pneumonia, thus CXR was justified at admission. The following line has been added in results

	 It is still not coming across clearly (at least to me) whether this system was one in which all children meeting IMCI pneumonia criteria had a CXR obtained or whether a CXR was obtained at the discretion of the treating physician. This is an important distinction. CXRs are not necessarily obtained on all children with clinical pneumonia, often only in children failing therapy or with more severe presentations, and so how these were obtained may introduce important bias into the study and should be addressed by the authors. 	section (page 14, line 233-234): “Among interpretable CXRs, 99.11 % (2804/2829) children had ‘severe pneumonia’ as per the WHO criteria¹⁹.” We have also listed all the signs of severe pneumonia in revised version of Table 1 (Page 15-17). The following signs have been added:  1. Oxygen Saturation < 90% 2. Grunting 3. Very Severe Chest Indrawing 4. Inability To Breastfeed Or Drink 5. Lethargy Or Reduced Level Of Consciousness 6. Convulsions 7. Central cyanosis
6.	Methods: CXR acquisition, page 8, line 100: When a hardcopy of the CXR could not be obtained from the caregiver after hospital discharge, please explain how the image of the printed film was captured and	The following lines have been added in the paragraph on page 8, line 103-105 “If the caregiver was not ready to give the hardcopy of the CXR (in <1% cases), image of the same was captured by surveillance officers using 16 megapixel cell phone camera and portable CXR viewbox.”

	whether a standard approach was applied. If not, this needs to be included as a limitation. Please also include in the results section how frequently this was an issue.	
7.	Methods: Page 11, lines 171-173: Please be specific regarding how the images were considered concordant or not.  • Was this only on PEP or no PEP? • Or was it considered discordant if one reader said PEP only and another said PEP with OI? 	The following has been added on Page 11, Line 173 and Page 14, line 238. In both places word “final conclusion” has been to the pre-existing line. “Interpretations were considered concordant when there was an agreement between two or more radiologists on final conclusion and discordant if all the three radiologists disagreed.”
8.	Methods:  • Page 13, line 212: Please explicitly define Model 1 • Please note Model 2 	The following has been added Model 1 has been explicitly defined under the section Interpretation of Radiological Images as follows (page 11, Line 166-168) as follows: “Final conclusions were ctegorised as: (a) “Abnormal” when it was `Primary End Point Pneumonia only` or `Other infiltrates only` or `Both PEP and other infiltrate` and (b) `Normal` when no findings were abnormal12.” Model II: Primary End Point Pneumonia (PEP) alone or with other infiltrates vs. Normal. The word “other” has been added now (page 13, line 213)

	should say "other infiltrates", currently says "infiltrate."	
9.	Methods: Given this analysis is based on medical chart extraction and the physicians documenting clinical signs were not trained for this study I would assume that missing data was not insubstantial. For example, physicians may not always document "no wheezing" if wheezing was not found on their examination. Please clarify how such a scenario was handled. Was it assumed by the authors that lack of documentation of a clinical sign indicated that the sign was absent? If this approach was taken did the authors do sensitivity analyses assuming that the data was missing and did this alter the findings?	Methods: The analysis is based on medical chart extraction by the surveillance officers. The study team did not assume that lack of documentation meant that a particular sign was absent. In case a clinical variable was missing from the medical chart, the surveillance officer contacted the clinician for obtaining information on the missing data. Thus, in clinical variables used for the analysis in the manuscript, there was no missing data.
10.	Results:	

	 • Page 14, line 228: I'm trying to understand why out of 3290 hospital cases screened the 3214 had WHO pneumonia? • How many children were hospitalized during this time period and how were these 3290 hospital cases identified for screening? I'm not following the surveillance system. • The fact that 3195 cases had CXRs suggests to me that CXRs were being obtained on nearly all children, regardless of clinical indication. Please clarify these points. 	 • Please refer to Figure 1 which shows that 3290 children were screened between Jan 2015-April 2017. Among these, 2.3 % (76/3290) refused to participate. Thus 3214 cases were included in the study. • We do not have information on the total number of children (2-59 months) hospitalized during this period. Among the hospitalized children of eligible age group , children with history of fast breathing with/without chest in-drawing were screened (n=3290) from hospital records for our project (Please see our screening criteria on page 6 line 48-50 which was also given in earlier version of manuscript). • All hospitalized children had WHO defined severe pneumonia. In our study, 99.1% (2804/2829) children were having severe pneumonia, according to WHO criteria¹⁹. Nearly all of them were therefore advised CXRs. The following line has been added in the `results` section (page 14, line 233-234). “Among interpretable CXRs, 99.11 % (2804/2829) children were having `severe pneumonia` as per the WHO criteria¹⁹.” We have also listed all the signs of severe pneumonia in revised version of Table 1 (Page 15-17). The following signs have been added:  1. Oxygen Saturation < 90% 2. Grunting 3. Very Severe Chest Indrawing 4. Inability To Breastfeed Or Drink 5. Lethargy Or Reduced Level Of Consciousness 6. Convulsions 7. Central cyanosis
11.	Results: Page 14, line 236: Clarify what is meant by concordance. Co	The following has been added on Page 11, Line 173 and Page 14, line 238. In both lines (line 174 and line 239) word “final conclusion” has been added to the pre-existing line “Interpretations were considered concordant when there was an agreement between two or more radiologists on final

	ncordance on what?  • PEP vs no PEP? • Or something else 	conclusion and discordant if all the three radiologists disagreed.”																					
12.	Results: Page 14, line 246: Why did the authors use <=90% for their hypoxemia threshold? The WHO uses <90% in the 2014 IMCI chart booklet and in the hospital pocketbook.	We thank the reviewer for pointing out this. This was an error that has been rectified now. Across the manuscript, we have now used WHO criteria of <90% oxygen saturation. Changes have been made at the following places  • Change in Table 1(Page 15-17).      Luck now Eta wah Pat na Darbh anga Tot al     Earlier manus cript Oxyge n Saturation ≤ 90 (%) 90 (17.04) 86 (26.7 9) 76 (32.2 0) 49 (15.36) 301 (20.5 8)   Revised manus cript Oxyge n Saturation <90% n (%) 61 (11.5 3) 57 (16. 61) 49 (20. 76) 43 (13.47) 210 (14. 72)     • Page 15, Line 250 • Page 6, line 57-58 			Luck now	Eta wah	Pat na	Darbh anga	Tot al	Earlier manus cript	Oxyge n Saturation ≤ 90 (%)	90 (17.04)	86 (26.7 9)	76 (32.2 0)	49 (15.36)	301 (20.5 8)	Revised manus cript	Oxyge n Saturation <90% n (%)	61 (11.5 3)	57 (16. 61)	49 (20. 76)	43 (13.47)	210 (14. 72)
		Luck now	Eta wah	Pat na	Darbh anga	Tot al																	
Earlier manus cript	Oxyge n Saturation ≤ 90 (%)	90 (17.04)	86 (26.7 9)	76 (32.2 0)	49 (15.36)	301 (20.5 8)																	
Revised manus cript	Oxyge n Saturation <90% n (%)	61 (11.5 3)	57 (16. 61)	49 (20. 76)	43 (13.47)	210 (14. 72)																	
13.	Results: Page 15 & 16, table 1: I'm not sure the p values have any meaning in this table? Consider removing?	As suggested by the reviewer, we have now removed p values from Table 1(Page 15-17).																					
14.	Table 1: Please add the clinical variables used to classify children as having pneumonia, including respiratory rate,	We thank the reviewer for this comment. As suggested by the reviewer, we have now added the following clinical variables in Table 1(Page 15-17).  1. Respiratory rate 2. Oxygen Saturation < 90% 3. Grunting 4. Very Severe Chest Indrawing 																					

	chest indrawing, WHO general danger signs. Include missingness if necessary.	5. Inability To Breastfeed Or Drink 6. Lethargy Or Reduced Level Of Consciousness 7. Convulsions 8. Central cyanosis
9.	Table 2: I'm curious as to how there is no missing data of clinical variables given my understanding is that these data were extracted from the medical file. This would be unusual. Please clarify.	There was no missing value of clinical variables reported in this manuscript. Surveillance officer extracted the available data from the medical file. In case, a clinical variable was absent or was not filled in the medical file, surveillance officer met the physician treating the child and noted this missing value from him/her.
10.	Discussion: page 22, line 332: I'm not sure abbreviating SP is necessary, can write out. I also don't think that the WHO definition of PEP is intended to be considered a surrogate marker of streptococcal pneumoniae per say. PEP was intended to be a specific CXR endpoint enriched with bacterial etiology, of which some % was streptococcal pneumoniae. For example, I would not suggest that 22% of CXR PEP means that 22% of children with CXRs have streptococcal	We have removed the abbreviation of SP at appropriate places  • Page 4, Line 11 • Page 27, Line 423 We agree with the reviewer. We have made the changes in wording at relevant places in revised manuscript (Page 23, line 332-333) "PEP can be taken as a good surrogate marker of bacterial pneumonia in epidemiological and vaccine efficacy studies"¹²

	pneumoniae. Your current wording suggests this is the case and this is not correct.	
11.	Discussion: Page 24, line 373: I would remove "possibly due to SP" from this sentence.	We have removed "possibly due to SP" from this sentence and at other places as follows: Page 25, line 385 Page 25, line 375

	Reviewer 2 : Kerry-Ann O'Grady Institution & Country: Queensland University of Technology Australia	
12.	Thank you for addressing the reviewer comments and the manuscript is much improved. I think you need to have a sentence or two in the discussion (limitations section) about what impact the lack of SpO2 readings, the proportion of children diagnosed with wheeze and the lack of information of prior antibiotic use actually has on the findings (eg, under-estimate, overestimate, is disease likely to be bacterial etc) rather than just saying it was a limitation, however this may need to be an Editor decision based on existing length of the manuscript.	We thank the reviewer for the comment. As suggested by the reviewer we have added few sentences in the discussion section as follows: Please see page 27 line 410-411. The following line has been added " This could also have lead to possibly over reporting of presence of wheezing. " Also see Page 27, line 416-417. The following lines have been added " Prior use of antibiotics could have possibly lead to underestimation of radiological pneumonia. We also observed that pulse oxymetry was routinely done in the network hospitals. This could have an impact on the case management but would not have affected the radiological findings of CXRs. "
13.	There still needs some work done on English grammar but it is not an impediment to eventual publication	As suggested by the reviewer, we have re-worked on the grammatical errors and tried our best to remove these, if any